# Numerical Simulation of the Transient Flow around the Combined Morphing Leading-Edge and Trailing-Edge Airfoil

**DOI:** 10.3390/biomimetics9020109

**Published:** 2024-02-12

**Authors:** Musavir Bashir, Mir Hossein Negahban, Ruxandra Mihaela Botez, Tony Wong

**Affiliations:** Research Laboratory in Active Controls, Avionics and Aeroservoelasticity (LARCASE), Department of Systems Engineering, École de Technologie Supérieure, 1100 Notre-Dame West, Montreal, QC H3C1K3, Canada; musavir-bashir.musavir-bashir.1@ens.etsmtl.ca (M.B.); mir-hossein.negahban-alvar.1@ens.etsmtl.ca (M.H.N.); tony.wong@etsmtl.ca (T.W.)

**Keywords:** morphing, unsteady parameterization, combined morphing leading edge and trailing edge (CoMpLETE), flow control

## Abstract

An integrated approach to active flow control is proposed by finding both the drooping leading edge and the morphing trailing edge for flow management. This strategy aims to manage flow separation control by utilizing the synergistic effects of both control mechanisms, which we call the combined morphing leading edge and trailing edge (CoMpLETE) technique. This design is inspired by a bionic porpoise nose and the flap movements of the cetacean species. The motion of this mechanism achieves a continuous, wave-like, variable airfoil camber. The dynamic motion of the airfoil’s upper and lower surface coordinates in response to unsteady conditions is achieved by combining the thickness-to-chord (*t*/*c*) distribution with the time-dependent camber line equation. A parameterization model was constructed to mimic the motion around the morphing airfoil at various deflection amplitudes at the stall angle of attack and morphing actuation start times. The mean properties and qualitative trends of the flow phenomena are captured by the transition SST (shear stress transport) model. The effectiveness of the dynamically morphing airfoil as a flow control approach is evaluated by obtaining flow field data, such as velocity streamlines, vorticity contours, and aerodynamic forces. Different cases are investigated for the CoMpLETE morphing airfoil, which evaluates the airfoil’s parameters, such as its morphing location, deflection amplitude, and morphing starting time. The morphing airfoil’s performance is analyzed to provide further insights into the dynamic lift and drag force variations at pre-defined deflection frequencies of 0.5 Hz, 1 Hz, and 2 Hz. The findings demonstrate that adjusting the airfoil camber reduces streamwise adverse pressure gradients, thus preventing significant flow separation. Although the trailing-edge deflection and its location along the chord influence the generation and separation of the leading-edge vortex (LEV), these results show that the combined effect of the morphing leading edge and trailing edge has the potential to mitigate flow separation. The morphing airfoil successfully contributes to the flow reattachment and significantly increases the maximum lift coefficient ($c_{l,max}$)). This work also broadens its focus to investigate the aerodynamic effects of a dynamically morphing leading and trailing edge, which seamlessly transitions along the side edges. The aerodynamic performance analysis is investigated across varying morphing frequencies, amplitudes, and actuation times.

## 1. Introduction

The increased demand for air travel has generated environmental concerns and significantly impacted the aircraft industry. The aviation sector aims for net-zero carbon emissions by 2050. To reduce CO_2_ emissions, the UN’s International Civil Aviation Organization (ICAO) organized negotiations and has given recommendations that include increasing aircraft innovation and “streamlining” flying operations. The aviation sector’s main objectives encompass enhancing air travel quality and affordability, reducing CO_2_, NOX, and noise emissions, and addressing safety concerns [1]. These challenges have spurred a greater need for innovative research to develop more efficient and environmentally friendly aircraft. The Research Laboratory in Active Controls, Avionics, and AeroServoElasticity (LARCASE) has continued to explore numerous strategies to reduce aircraft fuel consumption and emissions [2,3,4,5,6,7,8].

Advanced aerodynamic design technologies will play a crucial role in improving next-generation aircraft performance, reliability, and maneuverability. Flow separation control studies are critical to the aircraft industry [9]. At low Reynolds numbers, a laminar boundary layer forms on an airfoil’s upper (suction) surface, which is prone to separate with or without turbulent flow reattachment, depending on the angle of attack. When the angle of attack increases towards the stall angle of the airfoil, the adverse pressure gradient on the suction side increases. The flow separation point moves upstream and is displaced towards the leading edge. The adverse pressure gradient continues to increase to a certain point, at which the flow boundary layer separates near the leading edge on the suction side. As the angle of attack value attains the stall angle, the lift coefficient increases until it reaches the maximum lift coefficient. From that point, there is a significant drop in the airfoil lift and stall phenomena.

Various active and passive flow control technologies have been developed and applied in diverse practical applications [10,11,12,13,14,15,16]. Morphing technology can enhance an aircraft’s performance using various parameters, including lift, drag, and flow separation control [17,18,19,20,21]. During flight, mission requirements frequently change, and aircraft often operate in less-than-ideal conditions. In terms of aerodynamics, conventional aircraft employ hinged mechanisms for high-lift systems and trailing-edge surfaces to control the airflow. However, these mechanisms can also lead to increased drag [22]. These hinged surfaces have limitations when deployed and retracted [23]. Improper alignment of high-lift surfaces can generate noise and turbulence, resulting in flow separation.

Leading-edge morphing control is a widespread technique that facilitates alterations in wing curvature. The control of leading-edge vortices (LEVs) is achieved through active or passive modifications of the leading-edge shapes. The aerodynamic characteristic of the wavy leading-edge results in a gradual stall behavior characterized by a smooth reduction in lift rather than an abrupt loss. The concept of variable-droop leading-edge (VDLE) control involves the downward morphing of a specific section of the leading edge, wherein the morphing angle varies with time [24].

The variable-droop leading-edge (VDLE) design has demonstrated that the local angle of attack near it reduces when the overall angle of attack exceeds a certain threshold [24]. Consequently, this reduction mitigates the adverse pressure gradient, thus minimizing leading-edge vortex formation and the occurrence of flow separation and dynamic stall.

An investigation was conducted on implementing a combined leading-edge droop and Gurney flap to enhance the aerodynamic properties of both dynamic stall and post-stall in a rotor airfoil [25]. The experimental findings indicated that the onset of dynamic stall was delayed by 20 degrees when a leading-edge droop of 20 degrees and a Gurney flap with a chord length of 0.5% were employed. Another study employed VDLE using computational methods; their findings revealed a significant decrease in the drag and moment rise typically associated with dynamic stall [26].

However, it was also demonstrated that the drooped leading-edge system could not achieve the necessary lift and stall performance at high angles of attack compared to the advantages offered by a multi-element, typical high-lift system consisting of a slat and slot [27].

The aerodynamic performance of a morphing airfoil and wing with LE and TE parabolic flaps was quantitatively evaluated and further compared to prove its efficacy [28]. That work thoroughly analyzed airfoil aerodynamics using four typical design parameters. Parabolic flaps outperformed articulated flaps for airfoil lift increase of 39.2% and drag reduction of 108.4%, enhancing aerodynamic efficiency, respectively.

Variable-camber continuous trailing-edge flap (VCCTEF) technology was developed for a NASA general transport aircraft to study the drag reduction effect of the morphing TE [29], thus achieving an 8.4% drag reduction. Deflection frequency and angle effects were analyzed in a morphing LE airfoil’s aerodynamics under steady and unsteady conditions [30]. The impact of trailing-edge flaps (TEFs) on mitigating both the negative pitching moment and aerodynamic damping induced by a dynamic stall vortex (DSV) was examined in [31]. The authors have explained that these negative effects were linked to the trailing-edge vortex. Gerontakos and Lee conducted wind tunnel test studies to investigate the impact of TEF deflections on the dynamic loads generated by an oscillating airfoil motion [32]. The tests considered various parameters, such as the flap actuation start times, durations, and amplitudes. Interestingly, the results indicated that TEF’s motion did not affect the separation of the DSV. However, the duration of TEF deflection has impacted the airfoil’s maximum lift.

Unsteady flow characteristics and responses of a morphing airfoil under near-stall conditions were evaluated, and it was shown that a downward TE might increase lift instantly [33,34]. They analyzed the unstable aerodynamics of a morphing wing with a smooth side-edge transition [35]. Their results showed that the dynamically morphing wing’s unsteady flow impact might cause a substantial initial drag coefficient overshoot proportionally related to the morphing frequency. 

One approach to controlling dynamic stall involves using a leading-edge slat device, which was analyzed through numerical simulations using a two-dimensional Navier–Stokes solver for multi-element airfoils [36]. Another option consists of employing suction or blowing techniques. An experiment for dynamic-stall control was conducted using blowing slots positioned at 10% and 70% of the airfoil chord length [37].

One study demonstrated that the utilization of a small leading-edge angle (10°) morphing technique for a high angle of attack (>25°) leads to a notable enhancement in the efficiency of propulsion compared to that achieved with a rigid airfoil [38]. The work shown in [39] has demonstrated that the objective function determines the ideal suction speed. The approach involves modifying the airfoil’s geometry, and the airfoil’s nose radius was gradually adjusted to delay flow separation. At the same time, [40] used the nose droop of the VR-12 airfoil, setting the rotation angle at 25% of the chord and the droop angle at 13 degrees.

The potential of combustion-powered actuation to suppress dynamic stall on a high-lift rotorcraft airfoil was explored in [41]. Particle image velocimetry was employed to investigate the flow mechanisms involved in stall suppression. The test results indicated that combustion-powered actuators could significantly enhance the airfoil’s stall behavior at Mach number values up to 0.4. At Mach number 0.4, the cycle-averaged lift coefficient increased by 11% during the dynamic stall process. An investigation into the impact of a co-flow jet on the dynamic stall of a wind turbine airfoil was presented in [42]. The results showed that the dynamic stall could be significantly reduced, and energy consumption research revealed that the co-flow jet concept effectively controlled the dynamic stall.

Another study demonstrated that the transient lift coefficient associated with leading-edge (LE) morphing during the nose-up phase had a greater magnitude than that observed in the static situation [30]. The primary objective of the TE control is to enhance the moment characteristics and mitigate the dynamic stall compared to that achieved with conventional rudder surfaces. Performance enhancement techniques can be employed, such as trailing-edge (TE) morphing [43,44,45]. Combined LE–TE control integrates the characteristics of both LE and TE controls, such as the combination of LE droop, resulting in an enhanced suppression stall effect [46].

Another study explored the impact of a trailing-edge flap on dynamic stall effects at high speeds in wind turbines [47]. The pitching of the trailing-edge flap influences the dynamic stall hysteresis loops, which leads to load fluctuations. The consideration of the trailing-edge flap reduced the cyclic variability in the lift coefficient and the root bending moment by a minimum of 26% and 24%, respectively. These findings provide valuable insights into the advantages of trailing-edge flaps and their potential to mitigate load variations in wind turbine blades.

A numerical investigation focused on the blade section of a helicopter rotor during forward flight aimed to reduce or alleviate dynamic stall on the retreating blade [48]. For validation and verification purposes, simulations were conducted for flows around static and pitching airfoils at constant and variable Mach numbers. Their results were compared with other numerical data and experimental findings. Notably, the maximum drag coefficient ($c_{d,max}$) and the absolute value of the minimum pitching moment coefficient ($c_{m,min}$) were reduced by up to 49.2% and 25%, respectively. It is worth noting that the dynamic stall was eliminated using the proposed airfoil deformation under certain flow conditions.

The analysis focused on the leading-edge alteration inspired by cetacean species, specifically the porpoise [49]. This modification was applied to multiple airfoils, and the analysis was conducted at varied speeds. A three-dimensional numerical study was conducted on the impact of the spanwise placement and the extent of leading-edge alterations. The design of the porpoise nose, characterized by its shorter length and medium depth, has delayed flow separation and enhanced aerodynamic efficiency up to the crucial Mach number.

Recognizing the potential of both the drooping leading edge and the morphing trailing edge in flow control, this paper suggests an integrated approach to active control. This approach, called the combined morphing leading edge and trailing edge (CoMpLETE), seeks to harness the complementary effects of both control mechanisms to manage flow separation control effectively. Understanding the turbulent flow phenomena characterizing this approach will be valuable. Only a few articles in the literature thoroughly explore the combined leading and trailing-edge concepts [50,51,52], and only one study has explored parameters such as deflection frequency and magnitude and the starting morphing time [32].

This paper expands the scope to explore the unsteady aerodynamic consequences of applying CoMpLETE to an airfoil with a seamless flow transition. The study includes an aerodynamic performance analysis at different morphing frequencies, amplitudes, and actuation times. The objective is to understand how these diverse morphing frequencies affect the wing’s aerodynamic behavior and overall performance.

## 2. Morphing Model

The current research proposes to exploit the complementary effects of both leading-edge and trailing-edge control mechanisms to effectively manage flow separation control of an airfoil, using an approach called CoMpLETE (combined morphing leading edge and trailing edge). This design is inspired by the bionic porpoise nose and flap motions of the cetacean species. The motion of this mechanism achieves a continuous, wave-like, variable camber. The dynamic motion of the airfoil’s upper and lower surface coordinates in response to unsteady conditions is achieved by combining the thickness distribution with the time-dependent camber line equation. Additionally, the motion of the airfoil’s deflection is designed to start from its baseline configuration, progress to achieve a maximum downward target position, return to the original baseline configuration, and follow a sinusoidal pattern. Figure 1 depicts the model of various motions in cetacean species; it is evident that the mechanism undergoes slight oscillations, and similar airfoil motions can be obtained and investigated.

To obtain a fully dynamically morphing airfoil, precise definitions are essential for the changing shapes of the leading and trailing edges and their dynamic deformations. These definitions rely on a parametric approach that accurately represents the airfoil’s boundary and geometry. Incorporating control parameters derived from the four-digit NACA airfoil made it possible to dynamically modify the curvature of the camber line within the morphing section of the airfoil’s chord. This innovation allowed the design of a novel airfoil shape, incorporating time-varying elements into the parametric equations governing the trailing-edge geometry.

Because of the inherent asymmetry of the UAS-S45 airfoil, it was not feasible to directly apply or convert the mathematical model designed for symmetric airfoils. Consequently, a distinct approach was developed, explicitly tailored for asymmetric airfoils. In this context, the curvature and type of an asymmetric airfoil are determined by its camber. Figure 2 illustrates one of the morphing configurations, in which the leading-edge morphing starts at P, while considering the parameters for the leading- and trailing-edge morphing.

The dynamic motion of the airfoil’s upper and lower surface coordinates in response to unsteady conditions is achieved by combining the thickness distribution with the time-dependent camber line equation. Additionally, the motion of the airfoil’s deflection is initiated from its baseline configuration, progresses to achieve a maximum downward target position, returns to its baseline configuration, and follows a sinusoidal pattern. The following equation defines the parametrized time-dependent camber line:(1)$$\frac{y_{f}}{c}=\left\{\begin{array}{c} \frac{y_{c}}{c}-W_{le}\sin(2\pi tf)\frac{{\left(x_{i}-\bar{x}\right)}^{2}}{x_{i}^{2}},0\leq \bar{x}\leq x_{i} \end{array}\right\}$$
(2)$$y_{f}=\left\{\begin{array}{c} \;\;\;\;\;\;\;\;\;0,\;{\;\;\;\;\;x}_{i}\geq \bar{x} \end{array}\right\}$$
where $y_{f}\;$is the final y-coordinate of the new morphing airfoil camber line, $W_{le}$ is the value of the maximum deflection of the leading edge, and $\bar{x}$ is the x-coordinate of the selected control point, *t* is the time variable and *f* is the airfoil deflection frequency (cycles per second). The parameterization technique for the morphing trailing-edge motion is the same as that of the leading-edge motion. One of our previous papers explains the parameterization technique in detail [53]. Figure 3 depicts some of these parameters.

This research investigates the impact of this morphing approach on the flow separation occurring on the UAS-S45 airfoil. This study elucidates how the airfoil’s upward and downward deflections influence the separation bubble’s development during the airfoil’s oscillating motion.

Based on the morphing approach, different cases were set up with morphing motions (synchronous and asynchronous). Different morphing starting times and deflection frequencies were used to obtain different morphing shapes at different times. Two major studies were carried out based on the morphing airfoil chord location. In Table 1, the morphing of the leading edge starts at 0.2c, and the trailing-edge morphing starts at 0.75c; in Table 2, the morphing of the leading edge begins at 0.15c, and the trailing-edge morphing starts at 0.75c. The flow behavior was then analyzed for these cases, as shown in Table 1 and Table 2.

### 2.1. Computational Domain and Grid Definitions

The size and configuration of the computational domain significantly impact the accuracy of the results, especially concerning the aerodynamics of the geometry in question. In this particular aerodynamic scenario, the 2D computational domain forms a C-shaped grid, extending 20 times the airfoil chord length upstream (20c) and 30 times downstream (30c). These dimensions have been chosen to obtain a balance; the inlet and outlet distances are sufficiently large to ensure that the boundary conditions at the outer domain do not interfere with the airfoil’s flow, yet small enough to maintain reasonable computational efficiency.

This study used hybrid mesh, as depicted in Figure 4. These meshes combine structured quadrilateral layers near the airfoil with unstructured triangular elements elsewhere. Employing a blunt trailing edge makes designing smoothly transitioning O-shaped block layers surrounding the airfoil possible, thereby reducing instabilities that could arise from sharp corners at the trailing edge.

The wall condition parameter, denoted as $y^{+}$, serves as a dimensionless measure of the distance from the wall to the first grid point and is critical for assessing the “near-wall” mesh requirements. To accurately capture the characteristics of the near-wall region, a grid that aligns with the $y^{+}$ requirements of a chosen turbulence model is established. In this investigation, the ${\gamma -Re}_{\theta \;}$ turbulence model requires that the first cells near the airfoil should be positioned within the viscous sublayer, aiming for a $y^{+}$ value of less than one, which is recommended. The operating Reynolds number used was 2.4 × 10^6^ with a freestream velocity of U∞ = 34.5 m/s and a turbulence intensity of Tu = 0.1%.

### 2.2. Validation of Results

The validation methodology presented in our previous paper [45] was also used in this study. Comparisons are made with the existing data to ensure that the mesh(es) and physics are adequate to capture and provide consistent results of the flow characteristics. A convergence study was performed to verify that the mesh size was sufficient to capture flow details and obtain lift, drag, and moment coefficients for various mesh sizes. The mesh is varied by altering its base size, which scales the far field and refinement areas. It is worth noting that a maximum $y^{+}$ factor of 0.96 was identified along the airfoil surface during the second pitching cycle, thus indicating that the height of the first computational layer was correctly chosen to ensure accurate representation in its near-wall region.

The time –history variation of the lift coefficient illustrated in Figure 5a displays a sudden stall after the maximum value of the lift coefficient (peak).

Table 3 presents the characteristics of the three different grid sizes employed for their analysis. In comparison, Figure 6 illustrates the lift coefficient variations concerning the angle of attack for these three grid sizes. These results show a substantial degree of correlation in terms of lift coefficient.

For grid sizes one and two, the lift coefficients have slight differences in the downstroke phase, and there is a minor variation in their stall values (the peak of the lift coefficient curve). The lift coefficient fluctuations, shape, and magnitude also show remarkable similarity for all three grid sizes. During the upstroke phase, only very minor differences in the lift coefficients are found between grid size three and the other two grid sizes, while a small difference can be observed in the downstroke phase. It is important to note that the point at which the flow reattaches to the airfoil surface remains consistent for all three grid sizes. Consequently, grid size two was chosen as the computational domain due to its moderate cell number and overall good results.

Figure 7 presents a comparison between the computed lift and drag coefficients of the NACA 0012 airfoil model with their experimental values [54], as well as some of the numerical results from previous literature [55], under specific conditions of Reynolds number (2.5 × 10^6^), reduced frequency (k = 0.10), and angles of attack ranging from 5 to 25 degrees, with a mean incidence angle of 15 degrees. Two simulations were conducted: one was validated against experimental data, and the other was validated by comparison to numerical data from [55] for the same settings.

The ${\gamma -Re}_{\theta \;}$ turbulence model demonstrates the ability to capture the general trends in the results. It closely matches the experimental lift coefficient values during the upstroke phase but diverges when predicting the stall behavior. This discrepancy in stall prediction is attributed to the inherent dissipation in unsteady analyses, which reduces flow intensity and kinetic energy. Nonetheless, the numerical results align well with the trend of load variation up to the stall region, and differences in the peak values of the lift coefficient are relatively small. During the downstroke phase, variations arise due to the complex post-stall processes, leading to differences in the initial prediction of the LEV.

Figure 7b illustrates the variation of the drag coefficient with the angle of attack. A noticeable difference occurs at α > 10° and α > 12° in the case of numerical results in comparison to the experimental results, primarily because deep stall conditions cause a significant increase in drag. The drag coefficients from the upstroke [55] are lower than those at higher angles of attack (α > 22°) and lower angles of attack (α < 12°). However, the drag coefficients are higher than those obtained at angles of attack between 12° and 16°. However, the study’s numerical results indicate that the maximum drag coefficient of the airfoil is higher than that of the experimental values due to the formation of large vortices on the airfoil surface and the three-dimensionality of the flow. These vortices arise because of continuous flow separations at high angles of attack, making it challenging to accurately analyze the viscous effects near the airfoil surface. The CFD simulations in this study also reveal the presence of a secondary LEV that contributes to the recovery of the lift and drag coefficients around the maximum angle of attack.

## 3. Discussion of Results

Figure 8 shows the transient flow lift and drag coefficients of CoMpLETE airfoils at three cases for three frequencies of 0.5, 1, and 2 Hz at an angle of attack of 22° degrees (Table 1). This angle of attack was chosen because the flow was fully separated and its vortices were formed at 22°. For Cases 1.1, 1.2, and 1.3, the morphing of both leading and trailing edges begins at time *t* = 1.5 s and deflects synchronously in the same direction.

The average growth of these transient coefficients has been carried out by creating a baseline function of the transient data using the asymmetric least squares (ALS) smoothing method. The major peaks indicate that large changes in lift and drag coefficients are due to the vortex structure over the airfoil (Figure 8a,b). The period and amplitude of lift and drag fluctuations vary with frequency. Also, the lift and drag variations are not periodic, as seen over several cycles at each frequency, which demonstrates that vortex shedding has an aperiodic structure, and several vortices have various frequencies and intensities.

In Figure 8c, the airfoil morphing begins with time *t* = 1.5 s with a downward deflection at different frequencies for given cases in comparison to the baseline (not morphing) airfoil. The airfoils reach their maximum deflection at different scales due to frequency variations; the CoMpLETE airfoil in Case 1.1 reaches its maximum deflection at a time of 2 s, and the airfoil in Case 1.3 reaches a maximum deflection at 1.62 s. Therefore, the slope of the lift coefficient varies depending on the airfoil deflection frequency. At low frequencies, such as 0.5 Hz (Case 1.1), the lift coefficients’ peaks are higher as compared to those in other cases, and the average lift coefficient is also found to be higher than the other two cases. When the CoMpLETE airfoil starts to morph downwards, the vortex shedding decreases, which shows the reattachment of the flow to the airfoil. It is also seen that the high frequency leads to faster deflection and, therefore, sudden variations in the flow field. The average lift coefficient in Case 1.3 was found to be 1.58, which is lower than those obtained for the other two frequencies (Case 1.1 and Case 1.2), where the maximum average lift coefficient was 2.12 for Case 1.1.

The production and shedding of vortices over an airfoil affect its aerodynamics and create an unstable flow. However, time-domain curve analysis of the shedding vortex’s flow field features is complex. The discrete Fast Fourier transform (FFT) technique converts the aerodynamic coefficients from time to frequency. The amplitude spectrum was computed using the FFT algorithm based on the CoMpLETE airfoil’s transient lift coefficient. Figure 9 displays the lift coefficient’s power spectral distribution (PSD). There is a very small variation in magnitude. Nonetheless, several peaks are found in the data. First, vortex shedding over the airfoil causes tonal peaks. Also, large fluctuations in lift coefficients are associated with comparatively large lift variations near some frequencies. In addition, the energy content reduces as the frequency increases over time, which indicates that the vortices are reduced from larger to smaller structures. This also supports the findings above, that vortex shedding has an aperiodic structure and numerous vortices with various frequencies and comparable strengths over the large spectrum.

Three probes were placed around the airfoil to investigate the vortex shedding by measuring the local velocity component. One probe was positioned at 0.2c above the leading edge, and two were placed near the trailing edge in the wake. The velocity component at the leading-edge probe depicts the same phenomena shown in Figure 10. The peaks have more uniform values from 1.5 s to 2.5 s, as seen in Case (1.1), because the morphing occurs during this time. Similar behavior can be seen in Cases (1.2) and (1.3) in Figure 10. In these cases, the airfoil reaches maximum deflection faster due to a higher frequency than in Case (1.1). Therefore, uniform peaks are seen during the downward deflection, and as the flaps deflect upwards, the vortices emerge again. The cycle continues, and the lift and drag coefficients of the morphing airfoil follow the pattern.

The pressure coefficients of the CoMpLETE airfoil at the deflection frequencies of 0.5 Hz and 1 Hz (Case 1.1 and 1.2) at an angle of attack of 22° at different times are shown in Figure 11. Velocity streamlines and vorticity contours are also shown with these pressure coefficients. Figure 11a shows that at an angle of attack of 22° for the deflection frequency of 0.5 Hz, the pressure coefficient value reveals that the airfoil is already in a pre-stall condition at the time *t* = 1.45 s, and a large LEV is formed over the airfoil. The flow is characterized by a large vortex near the trailing edge of an airfoil, and the same phenomenon is observed at *t* = 1.5 s. The leading-edge and trailing-edge morphing starts at *t* = 1.5 s, and the airfoil starts to deflect downwards. At *t* = 1.62 s, the pressure coefficient shows that the flow slowly starts to reattach to the airfoil, and significant LEVs over the surface start to reduce. The velocity streamlines and vorticity contours reveal the same flow behavior.

In comparison to Case (1.1), Figure 11b shows a similar flow behavior for the deflection frequency of 1 Hz (Case (1.2)). However, the size of the flow separation areas over an airfoil varies because the deflection is higher at higher frequencies. The pressure coefficient at *t* = 1.64 s shows that the flow reattaches to the airfoil.

Figure 12a shows that at *t* = 2 s in Case (1.1), the morphing airfoil reaches maximum deflection, and the flow becomes stable. The leading and trailing edges start to deflect upwards to their original positions, and the pressure coefficient reveals that the flow remains attached at *t* = 2.4 s. Figure 12b shows the attached flow at *t* = 1.92 s; however, at *t* = 2.40 s, the deflection magnitude is higher due to the high deflection frequency of 1 Hz, and the airfoil deflects upwards from the baseline position, which results in the occurrence of its stall.

Therefore, these results reveal that the flow behavior changes with the deflection frequency and the extent to which the leading and trailing edges deflect at different times. In both cases, the leading and trailing edges deflect synchronously in the same direction.

Figure 13 and Figure 14 show the CoMpLETE airfoils with asynchronous morphing motion, where their morphing starting times for the leading and trailing edges differ. Cases (1.4) and Case (1.5) from Table 1 show the morphing starting time at a frequency of 0.5 Hz. Figure 13 shows an airfoil’s transient lift and drag coefficients for Case (1.4), where the leading edge deflects at 1.40 s and the trailing edge at 1.85 s at 0.5 Hz. Similarly, Figure 14 shows an airfoil’s transient lift and drag coefficients for Case (1.5), where the leading edge deflects at *t* = 1.85 s and the trailing edge at *t* = 1.4 s at 0.5 Hz. Therefore, there is a deflection time offset where the contribution of both the leading edge and trailing edge to the flow behavior can be observed.

This behavior is illustrated by comparing the lift coefficients of Cases (1.4) and (1.5) in Figure 15a. It can be seen that leading-edge morphing is more effective in stabilizing the flow over an airfoil by reducing the suddenly occurring pressure fluctuations. Figure 15b shows the instantaneous velocity variation with time. The velocity peaks at the leading-edge probe show stable behavior from time-step 1.5 s to 2.5 s in Case (1.4) and *t* = 2.4 s to *t* = 3 s in Case (1.5). The lift coefficient peaks shown in Case 1.4, where the leading edge deflects earlier than the trailing edge, are lower than those shown in Case 1.5; these coefficients are influenced by the deflection offset of the leading and trailing edges. The results indicate that the downward leading-edge motion significantly enhanced the post-stall lift situation by preventing DSV formation, resulting in fewer transient flow fluctuations. However, the trailing-edge deflection has a more significant impact on airfoil performance in terms of lift coefficient increase, as shown by the amplitude of the peaks. This is due to the vortex shedding and entrainment of these vortices on the pressure surface. These pressure variations result in multiple peaks that occur during vortices’ growth.

This behavior will be further explained by comparing the lift and drag coefficients of the synchronous airfoil with those of the asynchronous airfoil for the same deflection frequency, as shown in Figure 16. When the morphing starts in both Cases 1.1 and 1.4, the leading edge deflects downwards, and therefore, similar intensity peaks are seen, and the flow also becomes more stable compared to the time when there was no morphing. However, the trailing-edge deflection starts earlier in Case 1.5, and peaks with more intense fluctuations can be observed.

The underlying physics for the lift and drag coefficient fluctuations during morphing is due to the formation of the vortex structure. The energy stored by the shedding vortex progressively intensifies, forming a vortex structure on the suction surface. The maximal energy accumulation of the pressure surface’s vortex is shown by the peaks of the lift and drag coefficients. The increase in the lift and drag coefficient values predicts that the vortex will begin to move towards the trailing edge, creating a lifting force. When it reaches its peak, the separated vortex on the suction surface begins to fall off. Because there is no large-scale vortex structure on the suction surface at the time of leading-edge morphing, it presents a stable vortex structure, and, therefore, no intense fluctuation occurs in the lift at this time. The flow remains mainly attached to the airfoil and has no significant LEVs over the surface in downward deflection.

The above analysis provides insight into using leading-edge and trailing-edge deflections with different parameters. For example, the leading edge should deflect downward earlier, as slowly as possible, to make a stable vortex structure. The trailing edge should deflect less than the leading edge to improve the unsteady aerodynamic performance of the wings.

Evaluating the flow separation phenomenon requires tracking vortices’ formation, growth, and dissipation and determining their strength. Therefore, analyzing pressure distributions, velocity streamlines, and vorticity contours is significant. Figure 17a illustrates the pressure coefficients with velocity streamlines and vortices generated by the CoMpLETE airfoils for Cases (1.4) and (1.5) at a deflection frequency of 0.5 Hz and an angle of attack of 22°. In Case (1.4), when the morphing is initiated at *t* = 1.4 s, the low-pressure area’s peak has significantly decreased and is now spread across a larger surface. Different stall behaviors are the outcome of this changing load distribution.

In addition, a vortex is formed at the trailing edge, and the velocity streamlines are redirected upward toward the leading edge at *t* = 1.42 s. The leading edge continues to morph, and the flow re-attaches to the airfoil, increasing the negative pressure peak at *t* = 1.5 s and 1.53 s. In comparison, the trailing-edge morphing occurs earlier in Case (1.5), as shown in Figure 17b, and sudden pressure variations can be seen at *t* = 1.42 s. The flow remains detached from the airfoil, and very low pressure is spread across the airfoil, as seen at *t* = 1.5 s and *t* = 1.536 s. Figure 18a shows the sudden pressure fluctuations once the trailing edge starts to deflect at *t* = 2.21 s. The flow shows a sequence of vortices occurring on the airfoil surface at *t* = 2.31 s. When the airfoil’s leading edge deflects upwards, the trailing edge is still downward, and a sizeable leading-edge vortex can be seen at *t* = 2.54 *s*. Figure 18b shows the flow starts to re-attach with the airfoil when the leading edge deflects downwards and the trailing edge slowly returns to its original baseline position, as seen at *t* = 2.21 s and 2.31 s.

The effects of the morphing location of the leading edge on the flow behavior were investigated. Different cases were considered, as shown in Table 2. Firstly, Figure 19 shows the transient lift and drag coefficients for an airfoil with leading-edge morphing starting at 15%c (Case (2.1)) at the frequency of 0.5 Hz, and these coefficients show similar behavior as Figure 8’s transient lift and drag coefficients, where the leading edge also begins to morph at *t* = 1.5 s. There is a similar relationship between frequency and the period and amplitude of the variations in lift and drag. Similarly, the drag coefficient fluctuates as the leading and trailing edges morph. The abrupt pressure changes along the airfoil’s leading edge could cause these significant overshoots. Better aerodynamic efficiency is the outcome of the airfoil’s downward deflection.

Figure 20 compares Case (2.2), where leading-edge morphing begins at 15%c at the deflection frequency of 1 Hz, with Case (1.2), where leading-edge morphing begins at 20%c. Morphing starts at *t* = 1.5 s, and it can be seen that the maximum peak of the lift coefficient in Case (2.2) is higher than that of Case (1.2). The comparison of the leading-edge morphing locations of 0.2c and 0.15c showed that flow remains primarily attached to the airfoil in Case (1.2), and more intense velocity fluctuations are seen in Case (2.2). This is due to a changed leading-edge shape, with the velocity streamlines moving more intensely due to a sharp gradient in the Case of 0.15c, further lowering the local pressure.

Figure 21 shows the pressure coefficients, velocity streamlines, and vorticity contours for the same cases (Case (1.2) and Case (2.2)). The pressure fluctuations and vortices are shown for the CoMpLETE airfoil at a deflection frequency of 0.5 Hz at an angle of attack of 22°. It is shown that before morphing at *t* = 1.45 s, a series of vortices are observed over the airfoil, which may be expressed in terms of their respective pressure distributions. The camber of the airfoil changes as the morphing starts at *t* = 1.5 s. At *t* = 1.59 s, the airfoil is still in stall mode. Finally, at *t* = 1.64 s, the pressure distribution graph reveals that a maximum negative pressure arises, and the vortex moves towards the trailing edge. Finally, the flow reattaches to the airfoil at *t* = 1.924 s.

It can be seen that flow behavior remained approximately the same for Figure 21b, except at *t* = 1.92 s, where the flow is attached to the airfoil in Case (1.2), whereas flow is detached in Case (2.2).

Figure 22 and Figure 23 show the comparison of asynchronous morphing for different morphing location cases. As mentioned, Figure 22a compares the lift and Figure 22b drag coefficients as shown in Case (2.4), where leading-edge morphing begins at 15%c at the deflection frequency of 1 Hz, with those of Case (1.4), where leading-edge morphing begins at 20%c. The transient lift and drag coefficients show approximately similar behavior. Only slight differences in the strength of the peaks are seen. Figure 23 shows a similar comparison for Case (1.5) and Case (2.5). Figure 24 shows the same cases’ chordwise pressure distribution, velocity streamlines, and vorticity contours. In Figure 24a, the leading-edge morphs at *t* = 1.4 s; the pressure fluctuations and LEVs can be observed over the airfoil at (*t* = 1.42 s). At *t* = 1.53 s, the flow is reattached to the airfoil. In Figure 24b, a similar flow behavior is observed.

## 4. Conclusions

This paper investigated the effect of a combined morphing leading edge and trailing edge (CoMpLETE) on a morphing airfoil’s flow structure and behavior. This study focused on (1) the parameterization method for the morphing airfoil with LE and TE parabolic flaps and the influence of several design parameters: the LE droop angle, LE droop position, TE deflection angle, and TE deflection position, on the airfoil aerodynamics; and (2) the aerodynamic performance analysis for controlling the flow separation phenomenon. Various parameters, such as the airfoil’s droop nose amplitude and morphing starting time, were evaluated. The CoMpLETE airfoil performance was analyzed to provide further insights into the dynamic lift and drag force variations at pre-defined deflection frequencies of 0.5 Hz, 1 Hz, and 2 Hz. The lift and drag coefficients, velocity distribution, and vortex structure were analyzed in detail. The conclusions are the following:(a)Before the start of the airfoil morphing, an increase in lift coefficient is followed by a highly unsteady stall. A similar trend is shown for drag coefficients for all studied cases. The vortex structure, along with the velocity fluctuation, depicts this highly unsteady behavior.(b)As the morphing starts synchronously, the lift and drag coefficients of the CoMpLETE airfoil vary with the deflection magnitude and frequency. The maximum deflection is achieved at the time of 2 s in Case 1.1 and at 1.62 s in Case 1.3, respectively. Low deflection frequencies, such as 0.5 Hz give higher average lift coefficients and more stable flow, i.e., the average lift coefficient was 1.58 in Case 1.3, lower than that in Cases 1.1 and 1.2, where it was found to be 2.12.(c)The results also indicate that deflecting the trailing edge significantly impacted the airfoil performance in terms of lift coefficient increase. In comparison, the downward leading-edge motion significantly enhanced the post-stall lift situation by suppressing leading-edge separation and preventing the formation of the DSV.(d)The comparison of airfoil motion at the leading-edge morphing location at 0.2c and 0.15c showed that flow remains primarily attached to the airfoil in the case of the morphing location of 0.2c, and significant velocity fluctuations are seen in the case of morphing location of 0.15c. These results are due to a changed leading-edge shape, where the velocity streamlines move more intensely due to a sharper gradient in Case 0.15c, further lowering the local pressure.(e)The analysis of asynchronous CoMpLETE airfoil cases further validates the findings mentioned above. In the case where the trailing edge deflects earlier, higher lift coefficients can be obtained, with a more significant impact on airfoil performance. However, large velocity fluctuations were observed, and the DSV’s occurrence and detachment were not influenced. In comparison, the downward leading-edge motion significantly enhanced the post-stall lift situation by suppressing the leading-edge separation and preventing the formation of the DSV.(f)This asynchronous morphing motion needs further investigation because several parameters affect its performance, such as morphing start times, durations, and phase offsets.

A promising research direction will be to solve the flow fields of the morphing airfoil/wing using CoMpLETE airfoils with an oscillating motion to study the dynamic stall. The LARCASE’s Price –Païdoussis subsonic wind tunnel will be used for future wind tunnel studies. The findings are expected to clarify the flow physics and validate the findings of the unsteady flow behavior of the CoMpLETE airfoil.

## Figures and Tables

**Figure 1 biomimetics-09-00109-f001:**
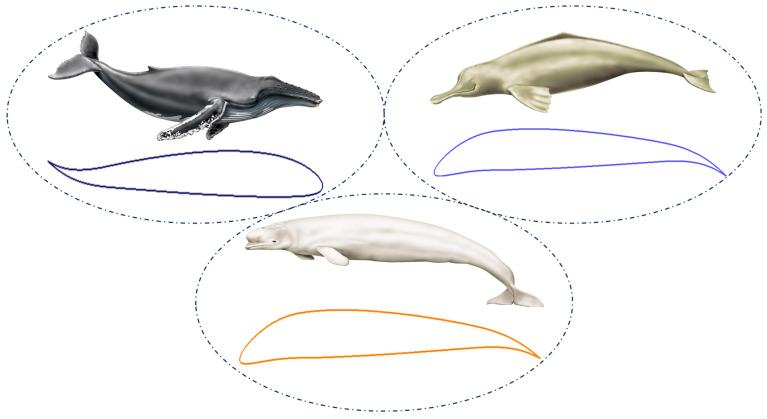
CoMpLETE (combined morphing leading edge and trailing edge) airfoils inspired by the cetacean species.

**Figure 2 biomimetics-09-00109-f002:**
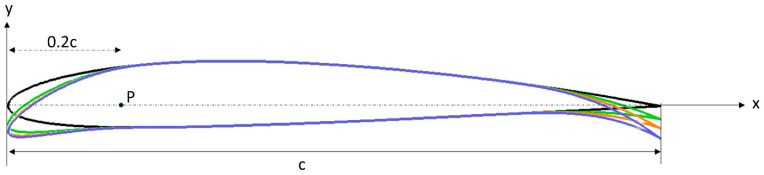
Geometrical definition of a variable camber line.

**Figure 3 biomimetics-09-00109-f003:**
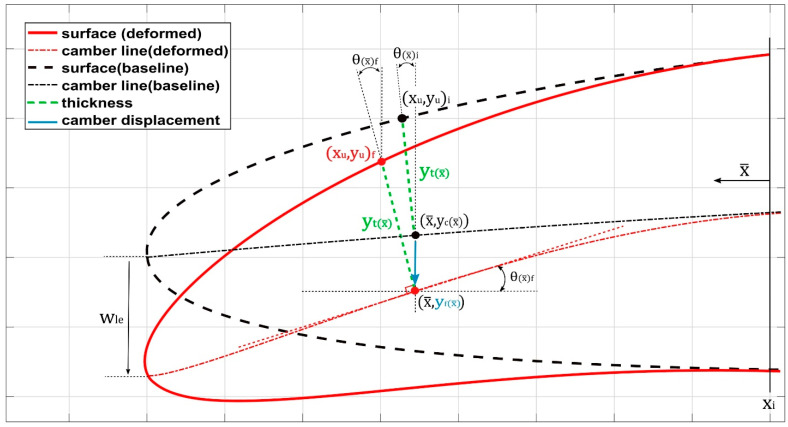
Numerical modeling of a camber line [53].

**Figure 4 biomimetics-09-00109-f004:**
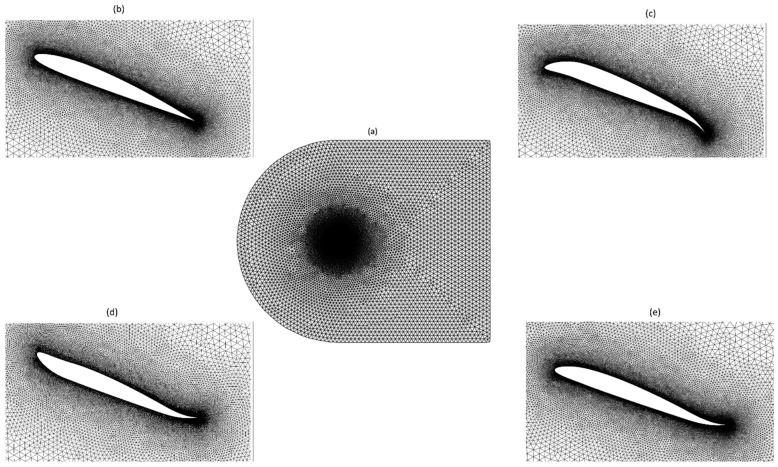
Computational domain with (**a**) overall mesh, (**b**) mesh around the reference airfoil, (**c**) mesh around the downward morphing airfoil, (**d**) mesh around the upward morphing airfoil, and (e) mesh around the asynchronous morphing airfoil.

**Figure 5 biomimetics-09-00109-f005:**
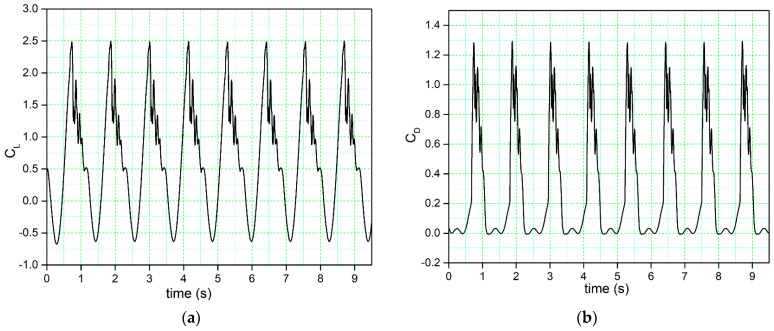
Time history of the (**a**) lift coefficient and (**b**) drag coefficient.

**Figure 6 biomimetics-09-00109-f006:**
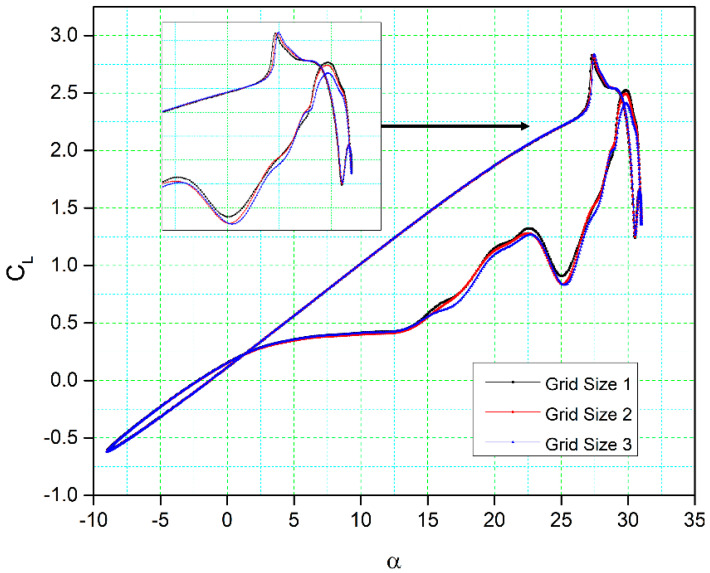
Comparisons of the numerical results for the lift coefficient versus the angle of attack for three different grid sizes.

**Figure 7 biomimetics-09-00109-f007:**
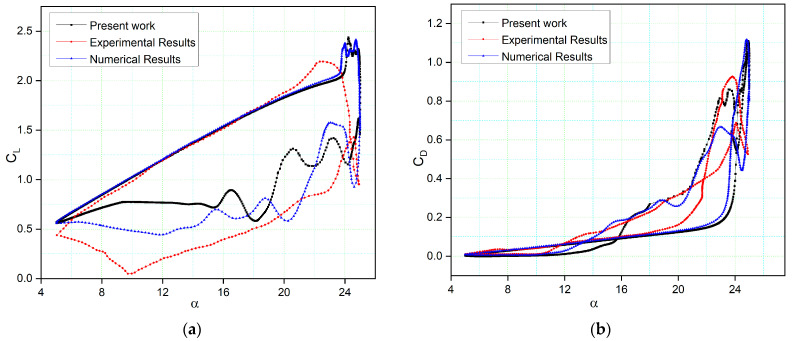
Comparison of our numerical results with experimental results obtained from wind tunnel tests [56] and numerical results [55]: (**a**) lift coefficient; (**b**) drag coefficient variations with the angle of attack.

**Figure 8 biomimetics-09-00109-f008:**
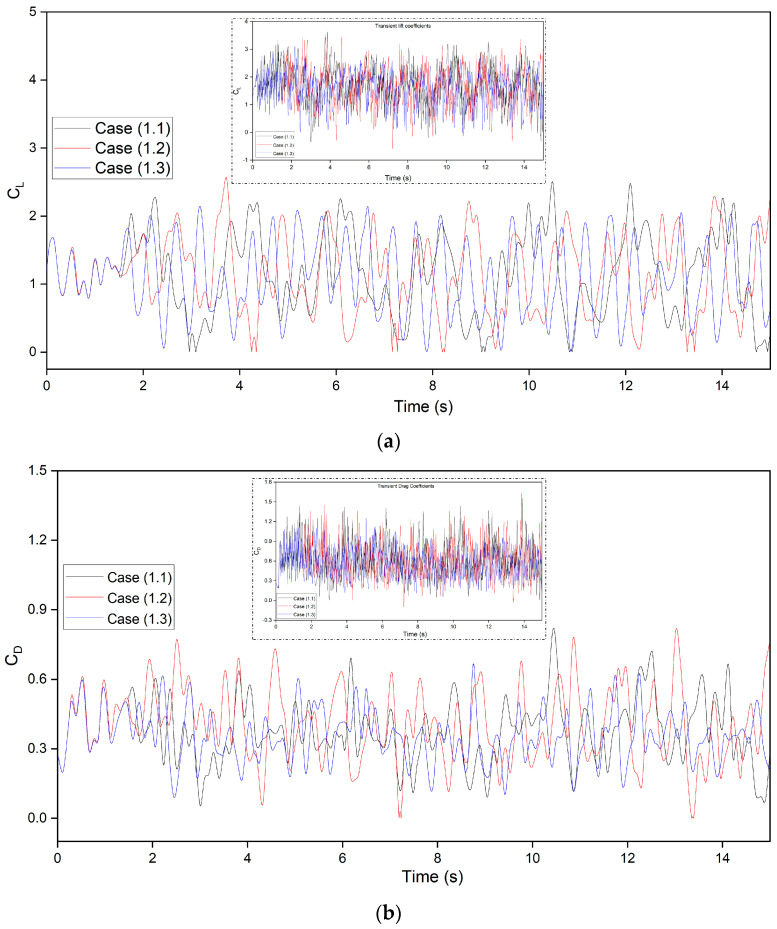
CoMpLETE airfoil for three different cases at 0.5 Hz, 1 Hz, and 2 Hz (**a**) shows an average growth of lift coefficients vs. time with the inserted figure of transient lift without baseline average, (**b**) shows an average growth of drag coefficients vs. time with the inserted figure of transient drag without baseline average, and (**c**) shows the lift coefficients for one cycle after the morphing starts at *t* = 1.5 s.

**Figure 9 biomimetics-09-00109-f009:**
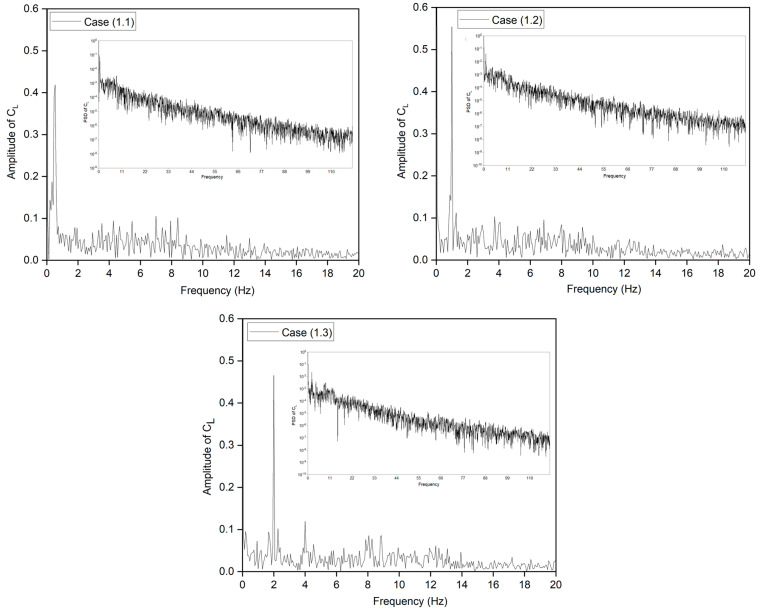
Power spectral density of lift coefficient for CoMpLETE airfoil for three different cases at 0.5 Hz, 1 Hz, and 2 Hz.

**Figure 10 biomimetics-09-00109-f010:**
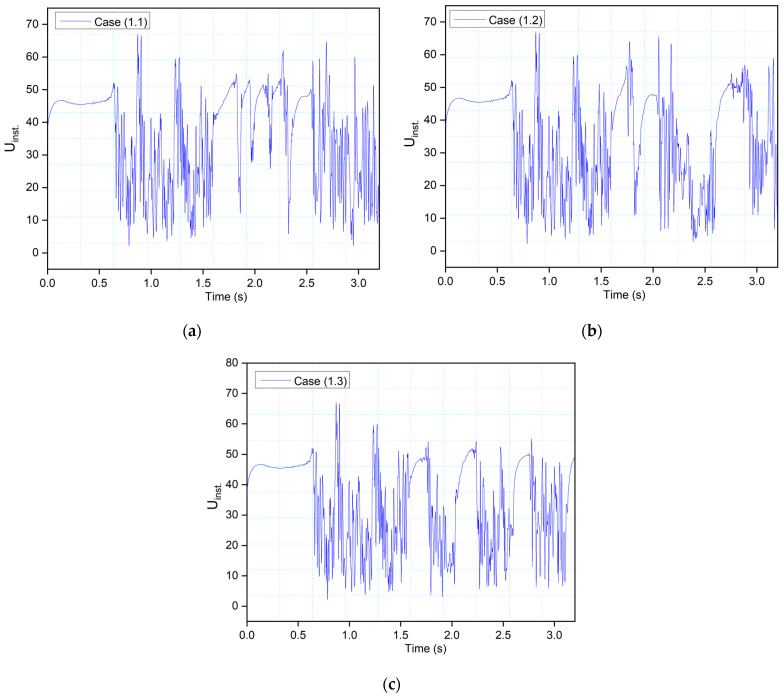
Local velocity component at the leading-edge probe position for the CoMpLETE airfoil at (**a**) 0.5 Hz, (**b**) 1 Hz, and (**c**) 2 Hz at an angle of attack of 22°.

**Figure 11 biomimetics-09-00109-f011:**
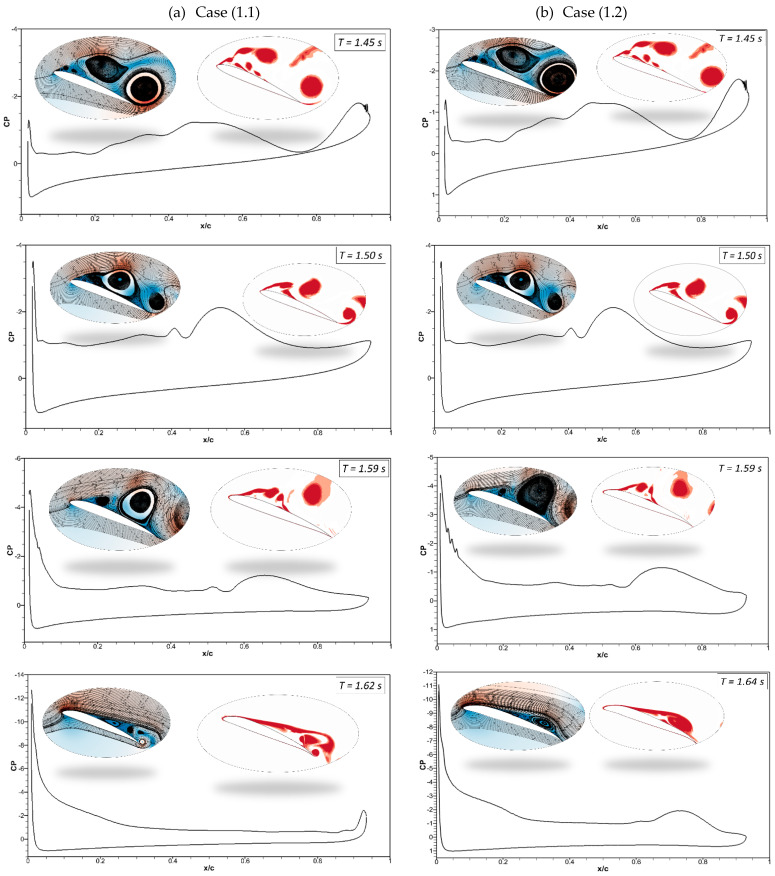
Pressure coefficients accompanied by velocity streamline and vorticity contours at different times for (**a**) Case ‘1.1’ and (**b**) Case ‘1.2’.

**Figure 12 biomimetics-09-00109-f012:**
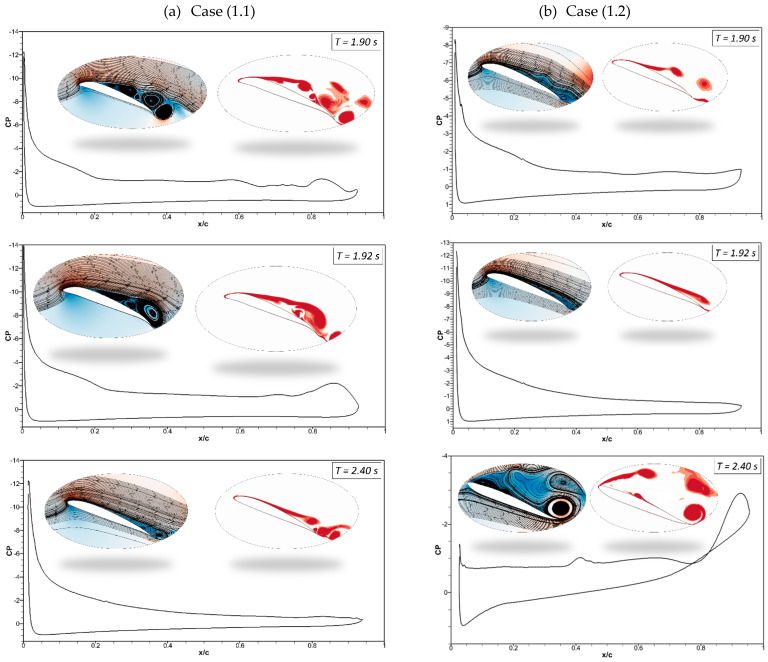
Pressure coefficients, velocity streamline, and vorticity contours at different times for (**a**) Case ‘1.1’ and (**b**) Case ‘1.2’.

**Figure 13 biomimetics-09-00109-f013:**
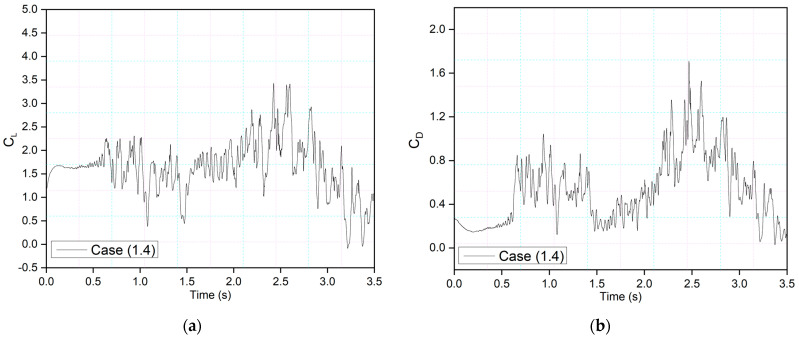
CoMpLETE airfoil for Cases 1.4 with (**a**) lift and (**b**) drag coefficient transient responses, at an angle of attack of 22°.

**Figure 14 biomimetics-09-00109-f014:**
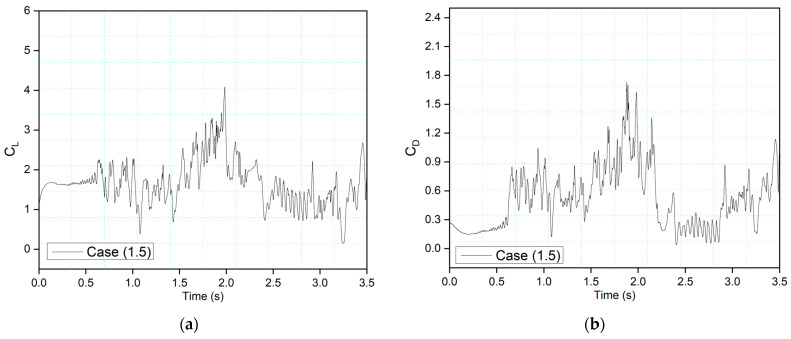
CoMpLETE airfoil for Cases 1.5 with (**a**) lift and (**b**) drag coefficient transient responses at an angle of attack of 22°.

**Figure 15 biomimetics-09-00109-f015:**
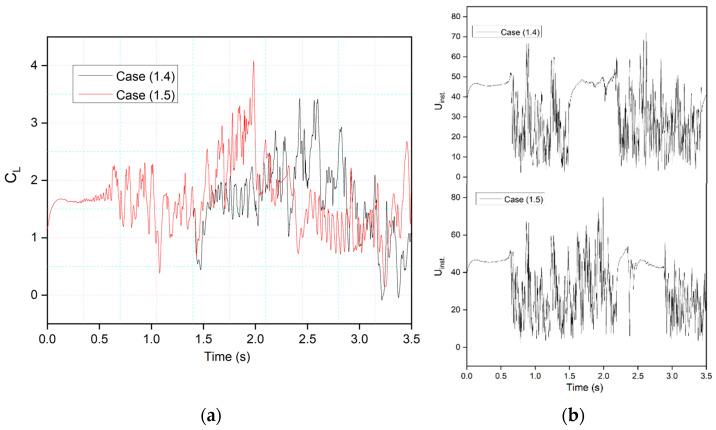
Comparison of (**a**) transient lift coefficient for Case (1.4) and (1.5) at 0.5 Hz. (**b**) Comparison of instantaneous velocity responses for Case (1.4) and (1.5) at 0.5 Hz.

**Figure 16 biomimetics-09-00109-f016:**
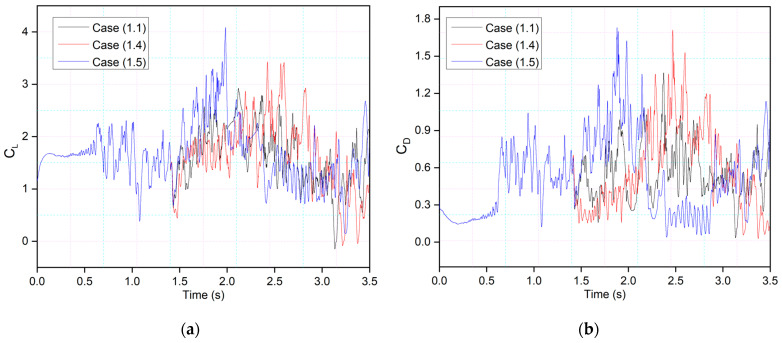
CoMpLETE airfoil for three different cases with (**a**) lift and (**b**) drag coefficient transient responses at an angle of attack of 22°.

**Figure 17 biomimetics-09-00109-f017:**
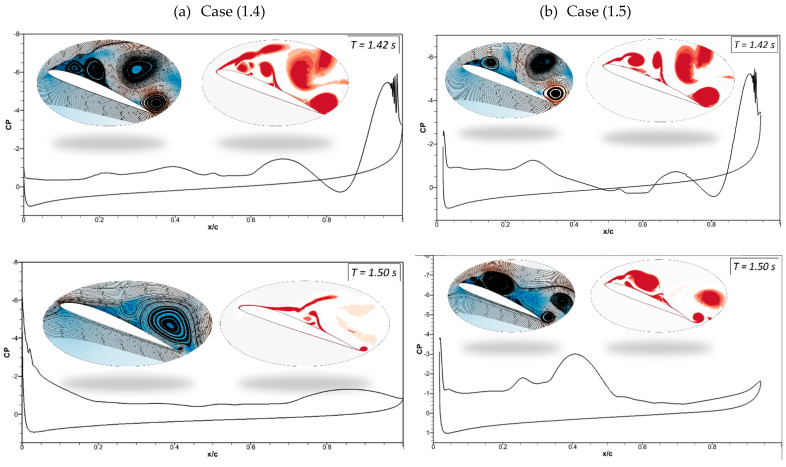
Pressure coefficients, velocity streamline, and vorticity contours at different times for (**a**) Case ‘1.4’ and (**b**) Case ‘1.5’.

**Figure 18 biomimetics-09-00109-f018:**
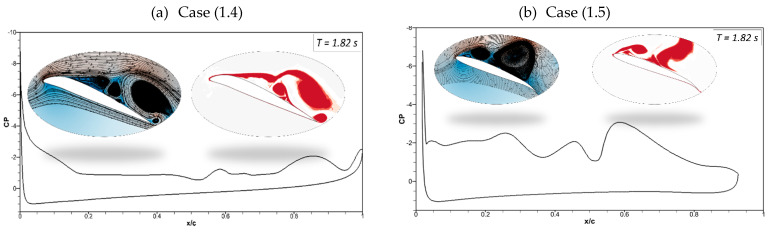
Pressure coefficients, velocity streamline, and vorticity contours at different times for (**a**) Case ‘1.4’ and (**b**) Case ‘1.5’.

**Figure 19 biomimetics-09-00109-f019:**
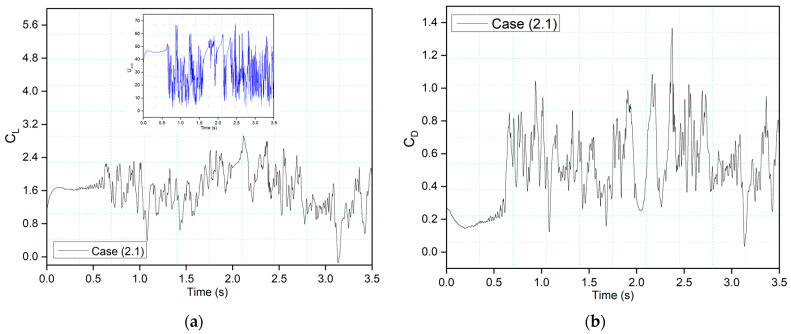
CoMpLETE airfoil for Case 2.1 with (**a**) lift and (**b**) drag coefficient transient responses at an angle of attack of 22°.

**Figure 20 biomimetics-09-00109-f020:**
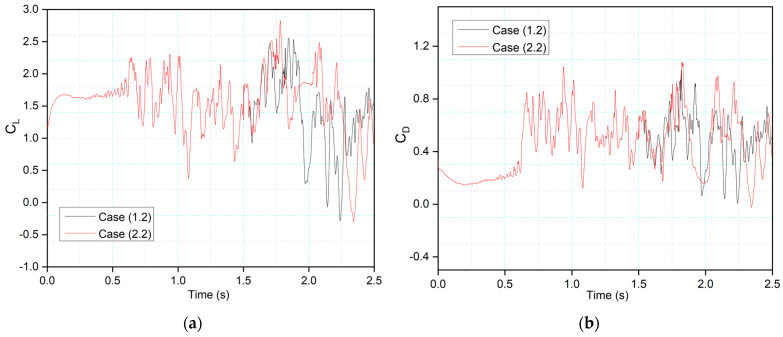
Comparison of transient (**a**) lift and (**b**) drag coefficient for Case 1.2 and Case 2.2 at 1 Hz.

**Figure 21 biomimetics-09-00109-f021:**
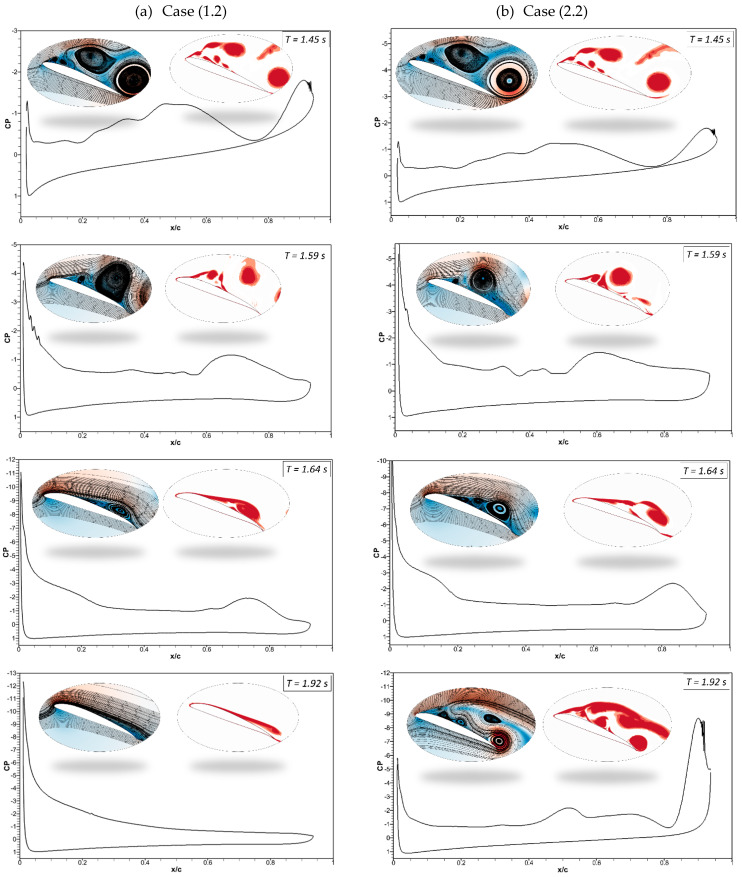
Pressure coefficients, velocity streamline, and vorticity contours at different times for (**a**) Case ‘1.2’ and (**b**) Case ‘2.2’.

**Figure 22 biomimetics-09-00109-f022:**
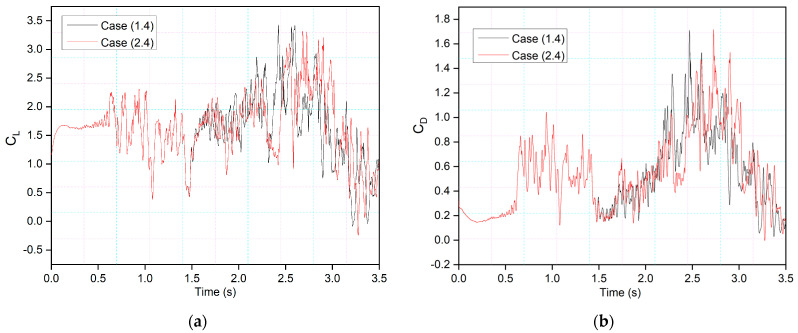
Comparison of transient (**a**) lift and (**b**) drag coefficients for Case 1.4 and Case 2.4 at 0.5 Hz.

**Figure 23 biomimetics-09-00109-f023:**
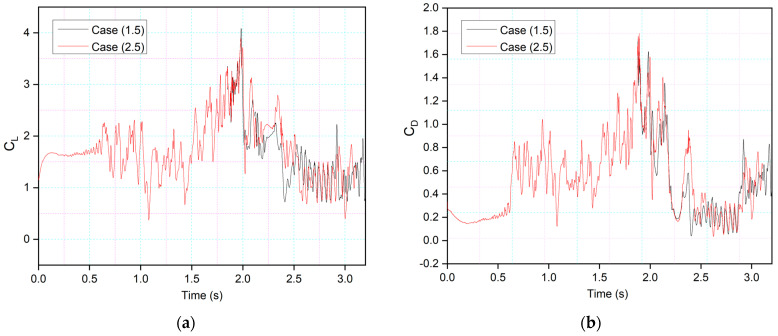
Comparison of transient (**a**) lift and (**b**) drag coefficients for Case 1.5 and Case 2.5 at 0.5 Hz.

**Figure 24 biomimetics-09-00109-f024:**
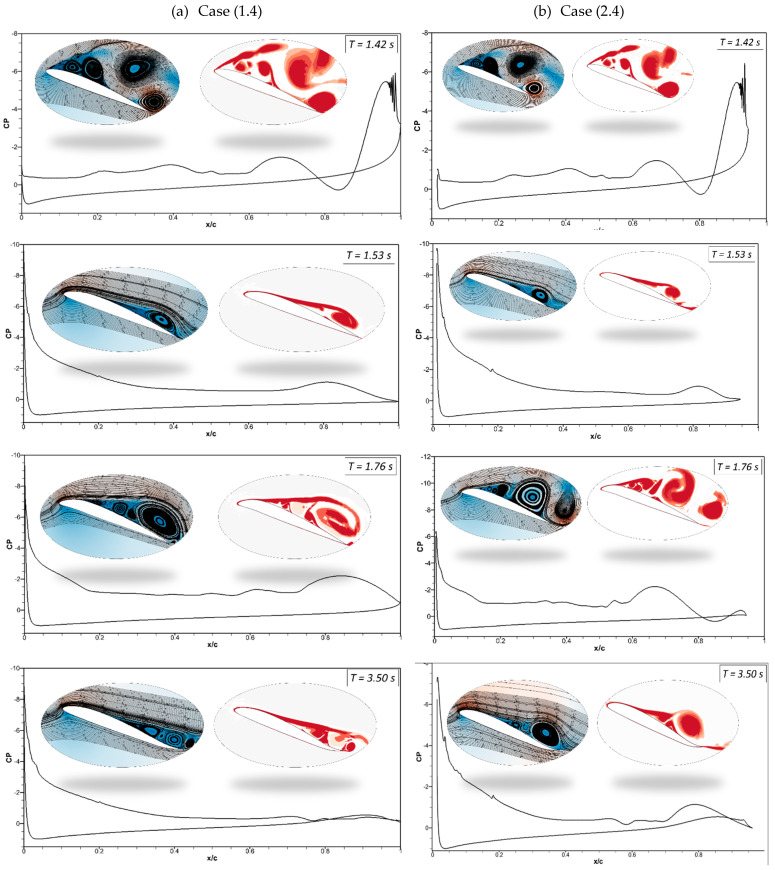
Pressure coefficients, velocity streamline, and vorticity contours at different times for (**a**) Case ‘1.4’ and (**b**) Case ‘2.4’.

**Table 1 biomimetics-09-00109-t001:** Airfoil with morphing of the leading edge starts at 0.2c, and the trailing-edge morphing starts at 0.75c.

Morphing Case	Leading-Edge Morphing Starting Time (s)	Trailing-Edge Morphing Starting Time (s)	Morphing Deflection Frequency (Hz)
Case 1.1	*t* = 1.5	*t* = 1.5	0.5
Case 1.2	*t* = 1.5	*t* = 1.5	1
Case 1.3	*t* = 1.5	*t* = 1.5	2
Case 1.4	*t* = 1.4	*t* = 1.85	0.5
Case 1.5	*t* = 1.85	*t* = 1.4	0.5

**Table 2 biomimetics-09-00109-t002:** Airfoil with morphing of the leading edge starts at 0.15c, and the trailing-edge morphing starts at 0.75c.

Morphing Case	Leading-Edge Morphing Starting Time (s)	Trailing-Edge Morphing Starting Time (s)	Morphing Deflection Frequency (Hz)
Case 2.1	*t* = 1.5	*t* = 1.5	0.5
Case 2.2	*t* = 1.5	*t* = 1.5	1
Case 2.3	*t* = 1.5	*t* = 1.5	2
Case 2.4	*t* = 1.4	*t* = 1.85	0.5
Case 2.5	*t* = 1.85	*t* = 1.4	0.5

**Table 3 biomimetics-09-00109-t003:** Grid properties of the three grid sizes for the grid-sensitivity analysis.

Grid Size	Number of Cells	Min Length	Max Length	Bias Factor
1	62,626	0.001	0.06	1.12
2	103,212	0.001	0.035	1.08
3	206,038	0.001	0.02	1.05

## Data Availability

The data presented in this study are available on request from the corresponding author.

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
