# Peer review of "Numerical Simulation of the Transient Flow around the Combined Morphing Leading-Edge and Trailing-Edge Airfoil"

_biomimetics, 2024, doi:10.3390/biomimetics9020109_

Round 1
Reviewer 1 Report
Comments and Suggestions for Authors
Dear authors,
The authors vividly demonstrate the relationship between the combined effect of deformation leading and trailing edges and airflow separation, and reveals the aerodynamic effects of dynamic deformation leading and trailing edges seamlessly transitioning along the side edges.
1) The paper is innovative and clear, with detailed data;
2) The description of the relationship between Vortex's movement and stall is very accurate.
The reviewer has several unimportant questions:
1) In Figure2, why is P (0.2c) used as the exact starting point for deformation?
2) In Tables 1-2, the morphing positions (MPs) of the leading edge start at 0.2c and 0.15, respectively. Then, how about MP=0.18c, MP=0.13c, MP=0.10c? What I mean is whether there is a clear trend and pattern of numerical fluctuations regarding flow behavior?
3) In Figure6, CLs are demonstrated based on the three grid sizes. The difference between the three of them is very small. Should the author make a specific comparison in terms of software execution time? After all, the time attribute of the number of grids in finite elements is very important
4)In Figures 8a-b, Why are the lift and drag coefficient curves discrete? In theory, they should be continuous.
In summary, the reviewer believes that this is a paper that can be directly accepted. If the author considers the questions I proposed, the manuscript will be more perfect.
Author Response
Dear Reviewer, please find here attached our answers to your comments and many thanks for the revisions of this paper, the authors

Reviewer 2 Report
Comments and Suggestions for Authors
In this paper the authors perform numerical simulatons of a dynamically combined morphing leading edge and trainling edge wing profile for the UAS-S45 airfoil.
The morphing considered is that of a periodic oscillation of the leading edge and the trailing edge in two variants, a "synchronous" one, where the leading edge and the trailing edge are starting their movement at the same instant, and subsequently in "asynchronous" mode, where the two parts begin the movement at staggered times.
The relevant question is whether there is an advantage in stall conditions to operating with the oscillatory motion compared to the static one, and which of the situations covered in the article is the most advantageous.
The problem is interesting and the simulations are interesting too, the problem in my opinion is with the presentation of the results.
Here are my main concerns:
I. The authors focus on the peak values over time of the lift and drag coefficients, which given the high temporal variability are maintained for very short moments of time, or they look at average growth of the lift coefficient during the downward morphing phase alone. In my opinion it would be most significant to report synthetic values averaged in time over the entire morphing period, instead of values measured locally in time or on a reduced time interval. Also, only the first period has been considered. An average over many periods would help identify trends.
II. Overall the conclusions are too vague. Part of the work is dedicated to investigating what happens if the movement of the leading edge and trailing edge is not synchronous but this part of the work is not mentioned in the conclusion. In another part of the work, a comparison between different leading edge morphing starting points has been done (casess 2.1-2.5), but there is no mention of the relevant results in the conclusions.
III. The stabilising effect of the leading edge appears to be predominant over that of the trailing edge. So is there any benefit of letting the trailing edge morph? Since the combined effect produced by morphing the leading edge and trailing edge is the real novelty of this paper, I think it is important to make the comparison with the case of the fixed trailing edge.
IV. The quality of English is lacking in many places and some sentences have no clear meaning.
V. The quality of some figures is very low and for this reason some features commented in the text that should be visible in the figures are difficult to distinguish.
In my opinion, therefore, the approach and simulations are interesting, but the results need to be rewritten in depth and reported in a more clear, convincing and readable way.
More in detail I enumerate specific points:
1. In the paper it is explained how the leading edge is made to move but not how the trailing edge is made to move.
2. Check equation (7): y_f appears both on the l.h.s. and on the r.h.s..
3. line 179: "Only a few articles in the literature thoroughly explore the CoMpLETE concept and its various parameters, including deflection frequency and extent and the starting morphing time": Please provide references.
4. Table 1: the results of cases 1.6 and 1.7 have not been presented: they can be deleted from the table.
5. Table 2: the results of case 2.3. have not been presented: it can be deleted from the table.
6. line 226: the control parameters M and P are not defined.
7. line 272: I guess reference [45] should be changed to [48].
8. line 321: "noticeable difference emerges when alpha > 12 degrees,": actually, differences between the numerical results start indeed to emerge at alpha=12. But with the experimental results, consistent differences can already be seen starting from alpha>10 degrees.
9. line 322: "The numerical simulations [50] show a lower drag coefficient with angle of attack than the values obtained in our simulations.": what interval are the authors referring to? The values of [50] are lower than those of the present work for alpha>22, but higher for alpha between 12 and 16 degree
10. line 323: "However, the study's numerical results indicate that the maximum drag coefficient of the airfoil is lower than that of the experimental values": Instead I see numerical peak values greater than 1.1 while the experimental peak value is around 0.9.
11. Figure 8 is too confusing. The lines are broken and cannot be perceived well. The curves overlap in many parts. Perhaps it would be better to separate them into three distinct figures and taking better care of the graphic appearance. The quality of the graphs in figures 10, 13, 14, 15, 16, 19, 20, 22, 23 also needs to be improved.
12. line 340: "The slope with which the lift increases during the morphing motion is proportional to the deflection frequency, the higher the deflection frequency, the steeper the slope." From figure 8 we see that the lift coefficient is highly variable. So how was the slope calculated? To better aprreciate proportionality with the frequency, the values of the slope as a function of the frequency could be reported in a table.
13. line 342: "At low frequencies such as 0.5 Hz (Case 1.1), the lift coefficients' peaks are higher than those at higher frequencies": I can't see it because figure 8 is very confusing. It would be appropriate to report the numerical values in a table. It is also necessary to explain why the peak value is important, given that the lift coefficient is very oscillating and the peak value is maintained for a very short time. Furthermore, due to the high variability of the signal, the peak value could vary significantly in subsequent periods. So why focus on the first period?
14. line 344: "When the CoMpLETE airfoil starts to morph downwards, the peaks are smaller, which shows that the flow reattaches to the airfoil. When the airfoil starts to return to its reference shape, the lift slope starts to increase again.": The peaks are smaller compared to what? And where can we see that the lift slope starts to increase again when the airfoil starts to return to its reference shape? In general in figure 8 I see a very noisy signal. If there are general trends, such as increasing or decreasing slopes and/or peaks between different cases, they should be better highlighted, for example by considering statistics of the signals over many periods.
15. Figure 8b is not commented on in the text.
16. Figure 9: in which time window was the power spectral density calculated?
17. line 355: where is the leading-edge probe positioned?
18. Figure 10: Explain in the caption what U_inst refers to.
19. line 348: "It is evident that there is a very small variation in magnitude.": what is the subject? I see a factor of 10^4 between the largest and smallest values.
20. line 350: "First, vortex shedding over the airfoil causes tonal peaks.": what are the frequencies of the tonal peaks? in figure 9 one cannot perceive them. Perhaps zooming in on low frequencies can help, or reporting the numerical values of the tonal peaks in the text.
21. line 356: "The speed peaks have almost the same values from 1.5 s to 2.5 s, as seen in case (1.1), because the morphing occurs during this time. Similar behavior can be seen in cases (1.2) and (1.3) in Figure 10. As the flaps deflect upwards, there is more flow separation. The cycle continues, and the lift and drag coefficients of the morphing airfoil follow the same pattern.". This is not very clear. What I see is, for all the three cases, a more uniform U_inst during the downward deflection phase, compared to a more oscillatory signal during the upward phase (due presumably to flow separation). Is this what the authors mean?
22. line 385: "Figure 12 (a) shows that at t = 1.9 s in Case (1.1), the morphing airfoil reaches maximum deflection". Isn't the deflection actually maximum at t=2 for case 1.1? Is there an error in table 1? Furthermore, in figure 11 the two figures on the left and right at t=1.50 are different, but the morphing has not yet started and therefore I would expect to see them the same. Unless there are errors in the second column of table 1.
23. line 392: "during different time steps.": better to say "at different times".
24. line 408: "The lift coefficient peaks in Case (1.4), where the leading edge deflects earlier than the trailing is lower than in Case (1.5).": the only way to find meaning in this sentence is to reformulate it like this: "The lift coefficient peaks in Case (1.4), where the leading edge deflects earlier than the trailing edge, are lower than in Case (1.5).". Otherwise what is the meaning?
25. line 410: "Because of the constant change in the effective deflection magnitude, the influence of dynamic morphing is illustrated by lift and drag over time.". this sentence is incomprehensible.
26. line 421: "The leading edge started to morph at t = 1.85 s, compensated for the sudden pressure variations and helped the flow reattach to the airfoil.". Another sentence that should be rephrased.
27. line 428: t=2.3s should be corrected to t=3s?
28. line 434: "The leading edge starts to move upwards back to its original position while the pressure drops significantly, and large fluctuations can be seen between t = 2.5 s and 3.5 s": where can we see the pressure drop?
29. line 436: "The above analysis provides insight into using leading-edge and trailing-edge deflection. For example, the leading edge should deflect downward earlier as slowly as possible and upward as quickly as possible, and the trailing edge should deflect less than the leading edge in order to improve the unsteady aerodynamic performance of the wings.": this conclusion is interesting but it should be supported by the results. Please elaborate.
30. line 501: " it can be seen that the maximum peak of the lift coefficient in Case (2.2) is higher than that of Case (1.2).". Is the maximum peak really important, given that it is maintained for a very short time?
31. line 519: "and the flow stabilizes and reattaches with the airfoil (Fig 26(b)). There is no figure 26.
32. line 519: "The flow takes longer to reattach to the airfoil for the same time step". I am confused by the addition of "for the same time step": to which time step do the authors refer?
33. line 536: "In Figure 21(a)". Probably they refer to figure 24(a)?
34. line 539 "Figure 21(b)". Maybe Figure 24(b)?
35. In caption of figure 22 change Case 1.2 to Case 1.4 and Case 2.2 to Case 2.4.
36. In caption of figure 23 change Case 1.2 to Case 1.5 and Case 2.2 to Case 2.5.
37. line 573: " the lift slope decreases as the leading-edge morphing begins, until it reaches the maximum deflection at low deflection frequencies.": another sentence to rephrase.
38. line 575: "When the airfoil returns to its original position, the lift slope increases again.": I may have had an oversight, but where do I find it in the text?
39. line 576: "The leading edge deflects upwards, increasing the flow separation and larger lift fluctuations.": maybe it was meant "and producing larger lift fluctuations."?
40. Please specify the Reynolds number of the simulations of section 3.
Comments on the Quality of English Languagemy comments on the quality of English language are written in the suggestions for authors.
Author Response
Dear Reviewer, many thanks for your comments and please find our answers to your comments that helped us to improve the paper.

Round 2
Reviewer 2 Report
Comments and Suggestions for Authors
I recognize that the authors have done an important work of arranging the paper. In my opinion the manuscript has been sufficiently improved to warrant publication in Biomimetics. I list just a few of the points already raised that I would like the authors to further improve. I don't need to double check the reworked version.
A 6 :"The parameterization technique for the morphing trailing-edge motion is the same as that of the leading-edge motion (explained in the paper). One of our previous papers explains the parameterization technique in detail (Ref. 51).": Please mention in the paper that the parametrization technique for the morphing trailing-edge motion is the same as that of the leading-edge motion.
A 7: ok for the first line of eq. (1). But for the second line, shouldn't yf = 0 be applied when \overline{x} >= x_i?
A 16: Figure 8 is now much clearer. However, the caption should be more descriptive: indicate the difference between the drawn curve and the upper insert in figures (a) and (b); specify what is plotted in figure (c).
A 22: please specify in the paper the leading-edge probe position.
A 23: please specify in the caption of figure 10 that U_inst refers to the local velocity magnitude at the probe position.
A 28: ok for the correction. Elsewhere too the "time step" (or "time steps") occurs which I would correct to "time" (or "times").
A 36: I still read at line 555: "and the flow stabilizes and reattaches with the airfoil (Fig 26(b)).". But there is no figure 26.
Comments on the Quality of English LanguageEnglish is now ok with only minor editing required.
Author Response
Dear Reviewer, many thanks for your comments and please find here attached our answers to your comments, thank you very much once again, Ruxandra Botez
